



# MIS 5e sea-level proxies in the eastern Mediterranean coastal region

Barbara Mauz[1,2], Dorit Sivan[3], Ehud Galili[4]

[1]School of Environmental Sciences, University of Liverpool, Liverpool, L69 7ZT, UK
[2]Department of Geology and Geography, University of Salzburg, Salzburg, 5020, Austria
5 [3]Maritime Civilizations Department, L. Charney School of Marine Sciences, University of Haifa, Israel
[4]Zinman Institute of Archaeology, University of Haifa, 199 Aba-Khoushi Avenue, Mount Carmel, Haifa, 3498838, Israel

*Correspondence to*: Barbara Mauz (mauz@liverpool.ac.uk)

**Abstract.** Mediterranean 'raised beaches' were subject to Quaternary research since the early years of the 20[th] century. The uniqueness of a warm-loving molluscs fauna immigrating into the Mediterranean made the coastline a prime subject for 10 studying Quaternary sea-level changes. Today, we have a detailed picture of this historically important coastline characterised by tectonically dormant coastal zone alternating with zones that are subject to subsidence or uplift. As part of the Word Atlas of last interglacial shorelines (WALIS) database we compiled 21 MIS 5e proxies for the eastern Mediterranean area available at http://doi.org/10.5281/zenodo.4274178 (Israel; Sivan and Galili, 2020) and at http://doi.org/10.5281/zenodo.4283819 (Turkey, Egypt, Tunisia; Mauz, 2020). All these datapoints are sea-level indicators 15 of variable quality situated between -1±4 m and 7±2 m resulting in a reconstructed MIS 5e palaeo-sea level situated between -1±4 m and 13±10 m.

## 1 Introduction

The eastern Mediterranean area (Fig. 1) is a remain of the western Neotethys Ocean (Hafkenscheid and Spakman, 2006) which formed when the Indian Ocean gateway closed during the Miocene (Bialik et al., 2019). It contains the oldest oceanic 20 crust on earth (Granot, 2016) which is actively subducting beneath the Aegean Sea (Crete, Peleponnese peninsula) and the Ionian Sea (Calabria). The oceanic crust is part of the northeast moving African plate and the continental part of this plate is a passive continental margin. The coasts of the eastern Mediterranean are therefore situated on earth's crustal segments that are, on late Quaternary time scales, actively deformed, slowly deformed or dormant. Because the eastern Mediterranean has a long history of geoscientific work the actively deformed areas, such as the Gulf of Corinth, Crete and Cyprus are very well 25 studied (e.g., McPhee and Hinsbergen, 2019; McNeill et al., 2018). These studies used, amongst other features, the relatively well-preserved last interglacial (LIG) marine terraces. On the other side, the coast of the African passive continental margin received attention through Quaternary scientist who aimed, in the first instance at least (e.g., Gignoux 2013), to carry forward the biostratigraphy of the late Tertiary owing to the fauna-rich coastal deposits. Today, our understanding of the geodynamics (e.g., Nocquet, 2012) enables us to attribute the eastern Mediterranean coastal zones to large-scale geological



structures (Fig. 1). This in turn enables us to separate shoreline data generated to unravel tectonic processes from sea-level data generated to reconstruct the LIG sea level and the associated ice volumes, eustacy and related GIA processes.

In this paper we describe previously published sea-level indicators on the basis of standards first developed by van de Plassche (1986) and Shennan, (1986), recently compiled in Shennan et al. (2015) and implemented by WALIS (https://warmcoasts.eu/world-atlas.html). We thereby contribute to the WALIS database described in Rovere et al. (2020)

available open-access at: DOI 10.5281/zenodo.4274178 for Israel (Sivan and Galili, 2020) and DOI 10.5281/zenodo.4283819 for the remaining eastern Mediterranean (Mauz, 2020).

1.1 Literature overview

The overview follows the clockwise spatial arrangement of large-scale geological structures in the eastern Mediterranean (Fig. 1) starting in the north. For the sake of clarity, we use names of eastern Mediterranean sub-basins supported by names

of nations. In terms of time we review literature which studied the "Tyrrhenian", a Mediterranean chronostratigraphic stage of the late Pleistocene that broadly encompasses the MIS 5 stage (Gibbard and Cohen, 2008). Historically, the Tyrrhenian stage has been identified on the basis of the so-called "Senegalese" fauna (Gignoux, 1913) which is an assemblage of warm-loving, shallow marine molluscs originating from the tropical Atlantic with the gastropod *Strombus bubonius* being its leading fossil. The Senegalese fauna immigrated into the Mediterranean during MIS 5 and disappeared with average sea-

surface temperatures falling below 20°C (Sessa et al., 2013). Therefore, the assemblage disappeared most probably ~115 ka, but may have remained until ~80 ka in niches along the warm north African coast.

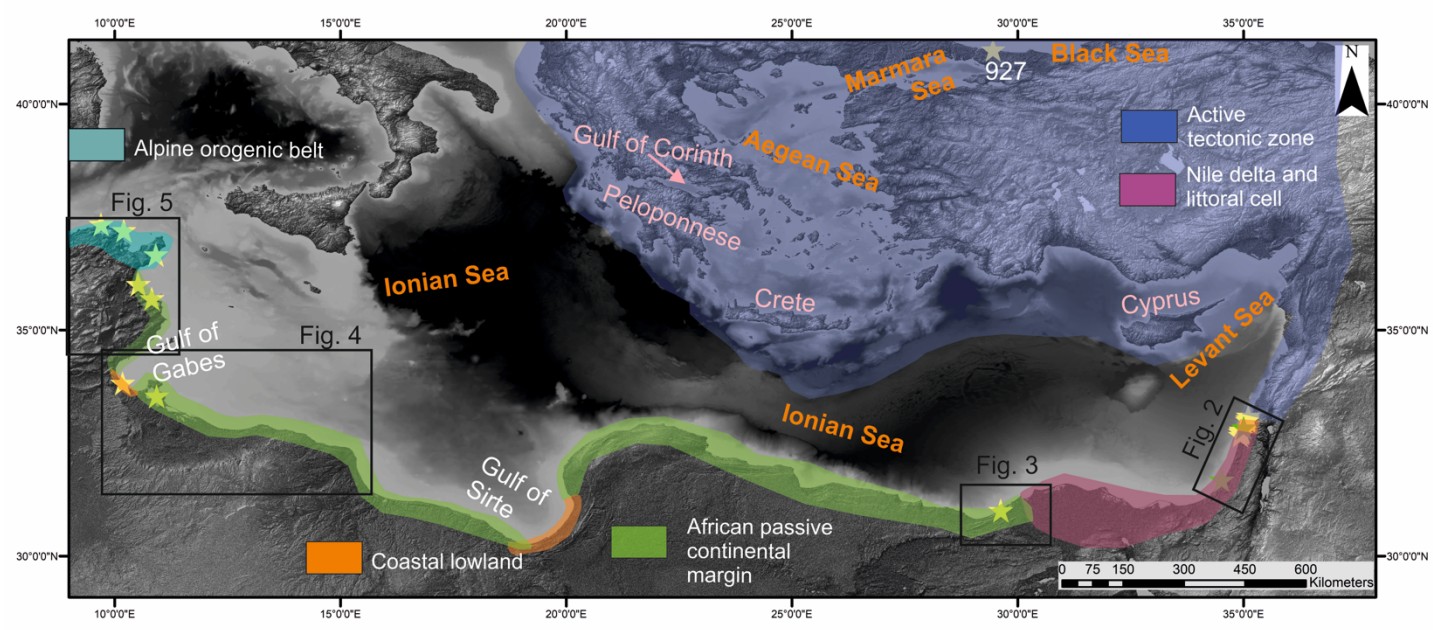



Fig. 1. The eastern Mediterranean and its coastal zones established for the purpose of this review. The zones are delimited according to the dominant geodynamic regime associated with the closure of the Tethys ocean (e.g., McPhee and van Hindsbergen, 2019), the African passive continental margin (Nocquet, 2012), the zone affected by the Nile discharge (Emery and Neev, 1960; Davis et al., 2012) and the morphological lowlands which are graben or rift basins with minor or absent tectonic activity during the late Quaternary. The Gebco data were obtained from GEBCO Compilation Group (2020) GEBCO 2020 Grid (doi:10.5285/a29c5465-b138-234d-e053-6c86abc040b9). SRTM data were obtained from http://srtm.csi.cgiar.org (Jarvis et al., 2008).

### 1.1.1  Active tectonic zone

*Ionian Sea (Peloponnese peninsula, Greece)* – this area is situated adjacent to the Hellenic trench and the Kephalonia fault both being part of the Hellenic subduction zone. Athanassis and Foutoulis (2013) identify MIS 5 shell-rich littoral deposits on a "platform" with a palaeo-shoreline situated at ca 40 - 60 m altitude.

*Ionian Sea (Corinth Gulf)* – the gulf is a marginal sea connected to the Ionian Sea through two 50-60 m deep sills. It is affected by high to very high extension rates and associated uplift of coastal zones. A suite of marine terraces occurs on the southeast coast of the Gulf (Vokha plain) which attracted many researchers owing to the large number of terrace levels (16 terraces; de Gelder et al., 2019), their well-preserved topographic expression and coastal sediment cover. The early studies focused on the relationship between sea-level highstands and terrace elevation (e.g., Keraudren and Sorel, 1987; Collier, 1990) followed by studies looking at the terraces, their sediment cover and associated ages and at structural implications with the seminal paper published by Armijo et al. (1996). The researchers' interest focused on the understanding of the Corinth rift (e.g., Gawthorpe et al., 2018) and we refer the reader to De Gelder et al. (2019) for the most recent study of the terraces and to McNeill et al. (2018) for the interplay between rifting, sediment routing, climate and sea level.

*Marmara Sea (Turkey)* – this is a marginal sea connected to the Aegean Sea through a ca 70 m deep sill (Dardanelles). Two major branches of the North Anatolian fault (NAF) delimit the Marmara basin to the north and south forming the Marmara transtensional basin (Jenkins et al., 2020). The northern NAF strand creates deep tectonic depressions separated by structural highs (Jenkins et al., 2020). High rates of NAF motion (~20 mm/a) progressively increasing westward (Bulkan et al., 2020) and associated earthquakes makes the area a prime subject of hazard studies. Studying Pleistocene marine terraces in this area aims therefore at determining uplift rates and, in addition, the hydrodynamic history between the Black Sea and the Aegean Sea. Yaltirak et al. (2002) studied the marine terraces along the Dardanelles coast and finds the LIG terrace at 38-22 m and at 9-0 m.

*Aegean Sea (Crete)* – the Crete island is a subaerial forearc above the Hellenic subduction zone and thus, its shoreline-related features are used to study the geodynamics of the eastern Mediterranean (e.g., Robertson et al., 2019). The studies focus on the south coast of the island where the LIG marine terrace is situated at 160-180 m altitude (Robertson et al., 2019) and at 50-100 m altitude (Gallen et al., 2014). For the most recent study of the terraces see Ott et al. (2019) and references therein.

*Levant Sea (Cyprus)* – the Cyprus island is situated above the Cyprean subduction zone. Similar to Crete, its marine terraces and related shoreline features are in the focus of structural geologists in order to understand the geodynamics of the eastern Mediterranean (e.g., McPhee and van Hinsbergen, 2019). Pleistocene deposits were studied by Vita-Finzi (1990; 1993),
Zomeni (2012), Poole et al. (1990) and Poole and Robertson, (1991) with the latter two studies showing that the "Tyrrhenian" terrace is situated at <3m on the south coast. On the north coast Galili et al. (2016) find the LIG terrace at 12-17 m while Palamakumbura et al. (2016) find the same terrace at 4 m altitude.

*Levant Sea (Turkey, Syria, Lebanon)* – the northernmost part of the Levant coast (Turkey) is part of the collision zone between the northward moving Arabian Plate, the westward moving Anatolian Plate and the southward moving African plate. There, Tari et al. (2018) studied marine terraces occurring within the Antakya Graben and find the MIS 5e marine
terrace at around 50 m altitude. Further south the coast is situated on the west flank of the sinistral Dead Sea transform fault characterised by coastal mountain ranges, pull-apart basins and graben (Brew et al., 2001). The coastal ranges are bounded by numerous normal and strike slip faults which are a consequence of the branching Dead Sea transform fault (Lebanese Restraining Bend; e.g., Weinberger et al., 2009). Dodonov et al. (2008) studied marine terraces on the Syrian coast and assigned the ones situated at 20 - 30 m to MIS 5. The southern boundary between deformed and dormant coastal zone is
represented by the Rosh Hanikra fault (Morhange et al., 2006).

*Ionian Sea (north Tunisia)* – the northernmost coast of Tunisia is part of the Alpine orogenic belt and associated tectonic processes. LIG deposits are part of cliff sections (Wided et al., 2019) or marine terraces. Elmejdoub and Jedoui (2009) show that in NE Tunisia the LIG deposits are part of a marine terrace at ~25 m altitude.

### 1.1.2 Nile littoral cell and Nile delta

*Levant Sea (Israel)* – The central ("Carmel") coast is part of the passive continental margin of the African plate which moves southward along the Dead Sea transform fault (e.g., Weinberger et al., 2009). It receives its sediments exclusively through the Nile littoral cell (e.g., Davis et al., 2012) which is a persistent easterly-driven longshore current created by the interplay of winds and Coriolis force. Coastal deposits bearing *Strombus bubonius* were first described by Issar and Kafri, (1972) and subsequently defined as "Yasaf" member by Sivan et al. (1999). LIG deposits occur on the Galilee coast at elevations 1-2 m
(Sivan et al., 2016), in Haifa bay at around -25 m (Fig. 2; Avnaim-Katav et al., 2012), on the Carmel coast at -1 m up to 7 m (Galili et al., 2007; 2018) and on the Sharon coast at around – 55 m (Fig. 2; Porat et al., 2003).

### 1.1.3 African passive continental margin

*Ionian Sea (Egypt)* – the coast sits on the African plate with minor to negligible effects from tectonic movements on the LIG deposits. The coastal plain west of the Nile delta exhibits a series of beach ridges first described and dated by El-Asmar 1994
and later by El-Asmar and Wood (2000) and studied by Elshazly et al. (2019). The second ridge behind the modern coastline was attributed to MIS 5e (Fig. 3).





*Ionian Sea* (*Libya*) – The coast from west of Alexandria (Egypt) to east of Tripoli (Libya) is under-studied and Quaternary deposits are only known from Explanatory Booklet provided in association to geological maps. Therein, Hinnawy and Cheshitev (1975) describe the "Gargaresh Formation" of "Tyrrhennian" age which forms an elongated ridge parallel to the
modern shoreline (Fig. 4).

*Ionian Sea (Tunisia)* –the coast is situated on the largest eastern Mediterranean shelf. Around Djerba island the LIG coastal environment is represented by a barrier stretching from Sabratah to the southern Gulf of Gabès (Fig. 4; Jedoui et al., 2003). A barrier also existed north of the Gabès gulf in the Gulf of Hammamet and along the south coast of Cap Bon. Coastal deposits were first systematically described by Paskoff and Sanaville (1983). Hearty (1986) provide the first age for the
Monastir site (Fig. 5) through his seminal work on amino acid razemisation correlated to U-series ages. Subsequently, Queslati (1994) and Jedoui et al. (2003) studied the LIG deposits on the south coast situated at 2-6 m altitude. Later, Mauz et al. (2009) studied sedimentological details of the deposits and their ages.

### 1.1.4 Coastal lowland

These coastal zones (Fig. 1) are morphological lowlands. Ongoing subsidence during the Quaternary is possible but not
everywhere evident from data.

*Black Sea (Turkey)* – this area is connected to the Aegean Sea through the ~100 m deep Bosporus sill. The Turkish Black Sea coast is in most places affected by tectonic uplift of up to 5 mm/a with the exception of the Sile site where several mm/a subsidence seems to prevail (Avsar et al., 2017). The GIA contribution to vertical land movements is estimated to ~0-1 mm/a (Avsar et al., 2017). Erginal et al. (2017) describes a shell-rich sandstone ("coquinite") at Sile situated within today's beach
zone and dated to around 128 ka.

*Haifa bay, Rosh Hanikra platform (Levant Sea, Israel)* - the northern ("Galilee") coast is situated between the Rosh Hanikra fault and the Carmel fault. The Ahihud fault separates the Rosh Hanikra platform from Haifa bay. Avnaim-Katav et al. (2012) report MIS 5e deposits from borehole data in Haifa bay at ca -25 m. No onshore coeval deposit is reported from this bay (Zviely et al., 2006).

*Gulf of Sirte* (*Libya*) – the gulf is part of a Cretaceous rift basin undergoing northward tilting and associated subsidence (van der Meer and Cloetingh, 1993). Giglia (1984) describes shallow marine oolitic limestone situated at around 3 m around 20 km inland.

*Gulf of Gabès* (*Tunisia*) – the gulf is part of the collapsed Jeffara block dominated by normal faulting during the Quaternary (Gharbi et al., 2016). Shape and geometry of the gulf funnels the tidal wave with the consequence that the gulf coast
experiences not only the highest tidal range in the Mediterranean (1.5 m; Gzam et al 2016) but also unusual shore-parallel hyrodynamic conditions (Gzam et al 2016). The LIG deposit is part of a beach ridge stretching parallel to the modern coastline at ca 3 m altitude (Fig. 4).



## 2 Sea-level indicators

Occurrence and preservation of sea-level indicators on the eastern Mediterranean coast is controlled by the morphology,
sediment supply and geological structure of the particular coastal zone as well as by the human demand for coastal resources.
Large coastal zones are deprived of indicators, e.g. the frequently cited zone at Monastir (Tunisia; e.g., Kopp et al., 2009).
On rocky, sediment-starved coasts erosional indicators such as abrasion platforms and tidal notches prevail while on soft
sediment coasts with sufficient sediment supply beach ridges, sandy beaches, barriers and spits prevail. Sediment-starved
coasts show abiotic carbonate crusts and lithophaga borings or biotic, reef-like constructions generated by red algae
(corallinacea), algal serpulids, coral (Cladocora caespitosa) or vermetids. Some of the indicators can provide small vertical
uncertainties, for instance, if the living range of a particular fauna is small or if the sediment facies is well constrained in
terms of water depth. For list of indicators see Table 1.

**Table 1: RSL indicators reviewed in this study.**

| Name of RSL indicator | Description of RSL indicator | Description of RWL | Description of IR | Indicator reference(s) |
|---|---|---|---|---|
| Marine terrace | Flat, gently seaward dipping rock surface bearing a veneer of coastal sediment and/or fauna | (Storm wave swash height + Breaking depth) / 2 | Storm wave swash height - Breaking depth | Rovere et al., 2016 |
| Coastal notch | Convex-shaped hole carved in hard rock | MSL | Tidal range = 30-40 cm | Antonioli et al 2015 |
| Sediment facies | Coastal conglomerate, siliciclastic or carbonate sand; oolitic sand, lagoonal silt and clay | MSL | backshore to shoreface (ca 0-10 m water depth) | Miall, 2010; Reading, 1986; Mauz et al. 2012 |
| Lithophagha cavity | Boring of genus *Lithophaga* in limestone | MSL | Living range: MSL to 25 m water depth | Coletti et al., 2020 |
| Cladocora caespitosa reef | Reef-like structure built by coral colony | MSL | Living range: 5-20 m water depth | Kruzic´ and Benkovic, (2008) |



| Petaloconchus vermid construction | Reef-like structure of sessile marine gastropod often in conjunction with encrusting algae and other organisms | MSL | Living range: 1-50 m water depth | Vescogni et al. 2008 |
|---|---|---|---|---|

## 2.1 Marine Terrace

The marine terrace is a gently seaward dipping surface typically covered by a veneer of coastal sediment. It typically occurs on uplifting (and subsiding) coasts where the interaction between wave action, hard-rock lithology and sediment starvation allow a wave abrasion platform to form during sea-level highstand. Because the inner margin, located updip of the terrace deposit, is often buried by terrestrial sediments, reconstruction of the maximum shoreline position is a challenge and requires digital elevation model (DEM) analyses (e.g. Jara-Muñoz et al., 2016). In the eastern Mediterranean many of the marine terraces occur around the Aegean Sea as a result of ongoing deformation of the lithosphere at the Hellenic subduction zone (e.g., islands of Crete and Cyprus), along the North Anatolian transform fault (e.g., Marmara Sea), along the Kephalonia transform fault (southern Balkans) and along extensional faults (e.g. Gulf of Corinth, Antakya Graben). Because the terraces bear testimony to late Quaternary deformation history, they are often investigated in tectonic studies where dimension and elevation are constrained through DEM, DGPS and similar techniques and the timing of formation is deduced from radiometric dating of the terrace surface and deposits. Thus, as long as additional techniques (e.g., GPR) are not employed, the marine terrace is considered marine-limiting or it is a sea-level indicator with an uncertainty deduced from the DEM resolution (e.g., 5 m).

## 2.2 Coastal Notch

Notches, typically carved in limestone cliffs, are a sea-level indicator formed by erosion. On the basis of form and shape they are differentiated in wave-cut notch and bio-erosional notch, both generated in the inter- to supratidal zone (Antonioli et al., 2015). Notches are regarded as accurate vertical markers of past sea level. When occurring on a micro-tidal coast, they are very precise because the vertical uncertainty is the tidal amplitude which is typically 15 - 30 cm in the eastern Mediterranean with the exception of the Gulf of Gabès (Tunisia) where the amplitude is around 70 cm. To use the notch as a sea-level indicator an unequivocally correlated and datable coastal deposit has to be available on site and this is rarely the case.

## 2.3 Sediment Facies

Eastern Mediterranean coastal sediments are represented by beach or fan-delta conglomerate, siliciclastic, carbonate and oolitic sand, clay and silt. These sediments may constitute beach ridges, barriers, veneers on terraces, beaches, lagoons or



sabkhas resting in coastal onlap architecture on a flooding or erosional surface. A coastal deposit is undifferentiated and, hence, marine-limiting if the depositional environment has not been determined. It is considered a sea-level indicator where facies determination allows inferring water depth. The indicative range of each sediment facies vary depending on the coastal topography, rate of sediment supply and prevailing hydrodynamics and must be determined for each site separately. In general, the indicative range of foreshore deposits is 1-4 m and of shoreface deposits it is 4-8 m water depth. With the

small tidal range in the eastern Mediterranean the vertical precision of an index point deduced from sediment facies can be <50 cm in places. However, with the difficulty of dating, the number of facies-based indicators remains small.

**2.4 Lithophagha Cavity**

The bivalve *Lithophaga lithophaga* belongs to the Mytilidae genera. Species of this genera are characterised by colonising hard substrates. They secret a calcium-binding substrate and thereby generate cavities that they inhabit (Coletti et al. 2020).

*L. lithophaga* live between the intertidal and 25 m water depth with highest abundance within 10 m water depth as reported for the Gulf of Naples (Coletti et al., 2020). The cavities are often observed on rock faces representing former cliffs and within the curvature of a notch suggesting that the bivalve colonises the cliff around mean sea level (Antonioli et al., 2015) with a relatively large uncertainty (>5m), however, owing to the large living range of the bivalve (Lambeck et al., 2004).

**2.5 *Cladocora Caespitosa* Reef**

*Cladocora caespitosa* is a scleractinian polyp characterised by a corallite skeleton. It is a colonial coral species which forms banks of variable size on rocky and, occasionally, on sandy seabeds. The temperate coral is endemic to the Mediterranean where it shows relatively slow skeletal growth rates ranging from 1.3 mm $yr^{-1}$ (Peirano et al., 1999) to 6.2 mm $yr^{-1}$ (Kružic and Požar-Domac, 2003) mainly controlled by water temperature (Rodolfo-Metalpa et al., 2008). The colonies grow in shallow water and are rarely found below 30 m water depth (Kružić and Benkovic, 2008) and are therefore considered

marine-limiting. Living and fossil *C. caespitosa* colonies are reported for north and central Aegean Sea, Marmara Sea, Crete, Cyprus, Levant, Gulf of Gabès, north Tunisia and Strait of Sicily (Ozalp and Alparslan, 2011). The coral has been used predominantly for the purpose of U-series dating the associated LIG deposits with limited success, however, likely caused by the strong seawater temperature dependence (Trotter et al., 2011) and associated crystal instability of carbonate minerals compared to their tropical counterparts.

**2.6 Vermetidae Reef**

The vermetid reef is a biogenic dome- or reef-like structure dominated by sessile marine gastropods living in shallow water of warm-temperate seas (Vescogni et al., 2008). Two species are known: Petaloconchus dominating the vermetids reefs in the Neogene until the Pleistocene, replaced by Dendropoma in the Holocene (Laborel, 1986). Vermetid reefs grow in associations with coralline algae, bryozoans, serpulids and benthic foraminifera creating together a bioclastic calcarenite

(Bosellini et al., 2001). While the living range of Dendropoma is well-constrained from modern analogues to be the intertidal



to upper subtidal zone (Laborel and Laborel-Deguen, 1994), the living range of Petaloconchus was reconstructed through facies analysis. These suggest beach down to the upper part of the slope (0-50 m; Vescogni et al., 2008) and, thus, Petaloconchus is considered marine-limiting.

**3 Elevation measurements**

Many of the studies listed above focused on stratigraphy, lithology or dating of the relevant coastal feature and did not emphasise elevation. Others, looking at the structural geology, reported elevation and associated measurement technique. Likewise, the sea-level datum was not reported to the detail desired for the WALIS database. In most cases "mean sea level" was used, also because the eastern Mediterranean tide is small and does not exceed 50 cm in most coastal zones. For the techniques used see Table 2.


**Table 2: Measurement techniques used to establish the elevation of MIS 5e shorelines.**

| Measurement technique | Description | Typical precision |
|---|---|---|
| TK Proflex 500 GPS | | 0.02 m to 0.08 m (Rovere et al., 2016) |
| Hand-held barometric altimeter | Difference in barometric pressure between a point of known elevation and a point of unknown elevation | Not reported |
| Topographic map | Contours | >10 m |
| Tape measure | Manually rolled tape measure and hand level | 50 cm |

**4 Eastern Mediterranean RSL sites**

In total 21 indicators are listed in the two databases and displayed against their longitudinal position in Fig. 6. When taking
the uncertainty into account, the minimum and maximum LIG sea level is at -5.1 m and 23 m, respectively. All facies-based indicators are characterised by small indicative ranges, but some are associated with large uncertainties owing to poorly constrained elevation data. From the 21 indicators displayed in Fig. 6 we describe here the 12 most reliable ones with their WALIS ID. These are situated on coastal zones affected by the interplay of eustacy and associated regional GIA only and



provide age constraints and/or the Senegal fauna. Zones with minor or debated non-GIA contributions are also included.

Coastal zones for which non-GIA processes during the late Quaternary are unambiguously evident are excluded (for zones see Fig. 1). For the selected zones we assume coastal geometries, hence tidal prism, to be similar to today because the course of the LIG beach ridges and barriers suggest a LIG coastline running sub-parallel to the modern one. The mean tidal range today is 0.5 m (Admiralty Tide tables) apart from the Gulf of Gabès (1.5 m; Gzam et al., 2016).

### 4.1 Black Sea rift zone

*Sile* (ID 927; Fig. 1) – the around 80 cm thick cemented, moderately sorted bioclast-rich sand ("Coquinite"; Erginal et al. 2017) shows seaward dipping laminae in its lower part and tabular planar cross-beds in its upper part; depositional environment is foreshore to beach (Erginal et al., 2017). The deposit is situated at 0±1 m. Assuming a virtual absence of tides in the Black Sea (Medvedev et al., 2016) and an operational Bosporus gateway during LIG, this is a facies-based sea-level indicator with an indicative range of 0-2 m water depth. The palaeo-sea level is at 0.0±1.5 m at 127±9 ka (Erginal et al.,

2017). The site is affected by subsidence estimated to ~2.3 mm/a (Avsar et al., 2017).

### 4.2 Nile littoral cell zone

*Nahel Me'arot* (ID 942; Fig. 2) – the around 5 m thick cemented oolitic grainstone shows seaward dipping planar laminae and the depositional environment is lower foreshore to upper shoreface deduced from modern analogue (Mauz et al., 2012). The elevation is 3-6 m (Galili et al., 2007). This is a facies-based sea-level indicator with an indicative range of 3-6 m water

depth. The palaeo-sea level is at 7±2 m at 113±5 ka.
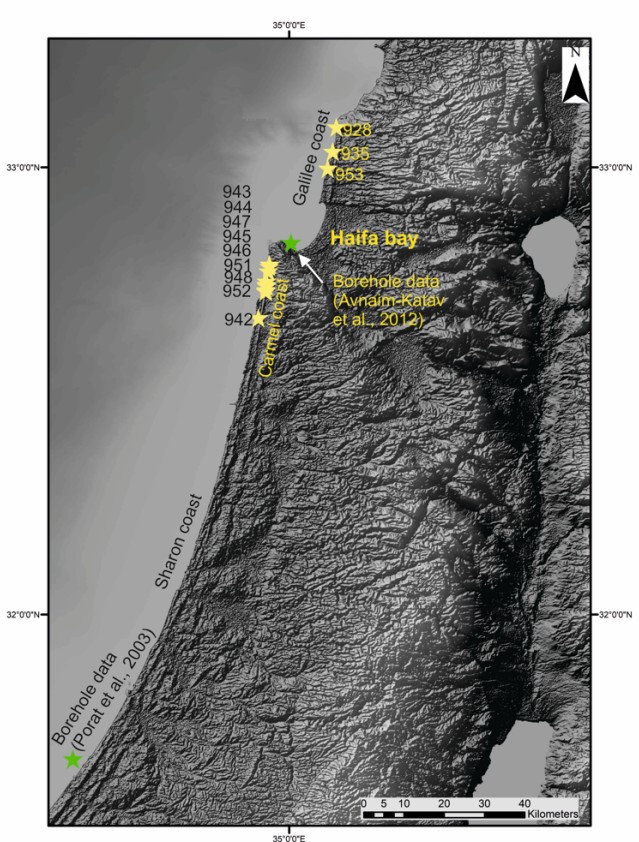

Fig. 2. The Levant Sea coast of Israel. Local names for the coast are Galilee, Carmel and Sharon for the northern, central and southern coast, respectively. Datapoints included in the WALIS database are depicted with their WALIS ID (e.g. 945). Green stars indicate location of borehole. Gebco data were obtained from GEBCO Compilation Group (2020) GEBCO 2020 Grid (doi:10.5285/a29c5465-b138-234d-e053-6c86abc040b9). SRTM data were obtained from http://srtm.csi.cgiar.org (Jarvis et al., 2008).

### 4.3 African passive continental margin zone

*El-Max-Abu Sir* (ID 1362; Fig. 3) – The LIG shoreline is represented by the second beach ridge behind the modern coastline. Cross-bedded bioclastic and oolitic grainstone of foreshore depositional environment (Elshazly et al., 2019) constitute the basal part of the ridge. The elevation should be < 5 m but is poorly defined. This is a facies-based sea-level indicator with an indicative range of 1-3 m water depth. The palaeo-sea level is at 0.5±4.6 m at 121±6 ka (El-Asmar, 1994).



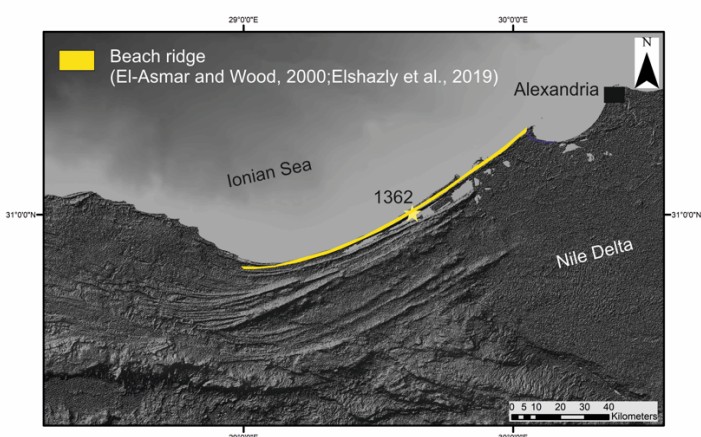

Fig. 3. The Ionian Sea coast west of the Nile delta (Egypt). The WALIS datapoint is ID 1362. Gebco data were obtained
from GEBCO Compilation Group (2020) GEBCO 2020 Grid (doi:10.5285/a29c5465-b138-234d-e053-6c86abc040b9).

SRTM data were obtained from http://srtm.csi.cgiar.org (Jarvis et al., 2008).

*Ras Karboub* (ID 1363; Fig. 4) – This site is part of the Jeffara barrier stretching almost parallel to the modern coastline from
Sabratah (Libya) to Djerba island (Tunisia). The barrier shows siliciclastic sand of shoreface to foreshore gradually passing
into oolitic grainstone of foreshore environment (Jedoui et al., 2003; Mauz et al., 2009). The elevation should be 5-10 m but

is poorly defined. This is a facies-based sea-level indicator with an indicative range of +1m to -3 m. The palaeo-sea level is
at 7±10 m at 114±16 ka (Mauz et al., 2012).

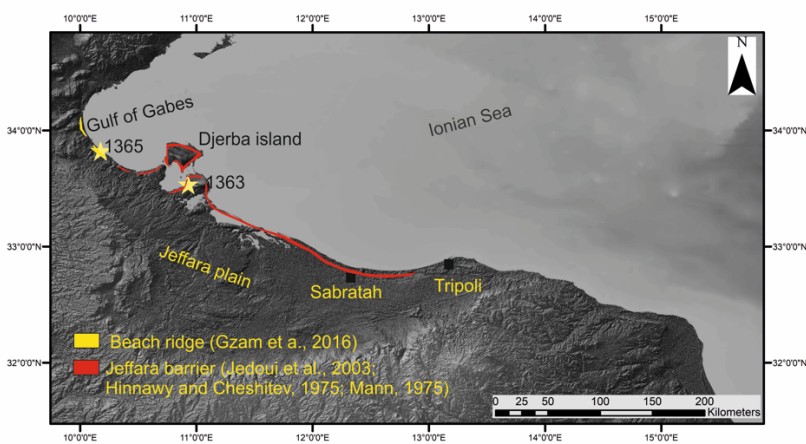

Fig. 4. The Ionian Sea coast between the Gulf of Sirte and the Gulf of Gabès (Libya and Tunisia) The datapoints included in
the WALIS database are depicted with their WALIS ID. Gebco data were obtained from GEBCO Compilation Group (2020)

GEBCO 2020 Grid (doi:10.5285/a29c5465-b138-234d-e053-6c86abc040b9). SRTM data were obtained from
       http://srtm.csi.cgiar.org (Jarvis et al., 2008).

       *Khniss* (ID 1364; Fig. 5) - This site is part of the Hammamet barrier stretching almost parallel to the modern coastline from
       Chebba to ID 926. The barrier shows siliciclastic sand of shoreface to foreshore environment gradually passing into oolitic
grainstone of foreshore environment (Jedoui et al., 2003; Mauz et al., 2009). The elevation should be 5-10 m but is poorly
       defined. This is a facies-based sea-level indicator with an indicative range of +1m to -3 m. The palaeo-sea level is at 10±3 m
       at 121±10 ka (Mauz et al., 2009).

       *Hergla* (ID 926; Fig. 5) - the around 2 m thick cemented bioclastic quartz sand shows planar laminae; the depositional
       environment is foreshore (Mauz et al., 2018). The elevation of the corresponding shoreline should be at 3-2 m deduced from
the altitude of the coeval lagoonal deposit (Mauz et al., 2018). This is a facies-based sea-level indicator with an indicative
       range of 1-3 m water depth. The palaeo-sea level is at 4.2±1.8 m at 120±5 ka.

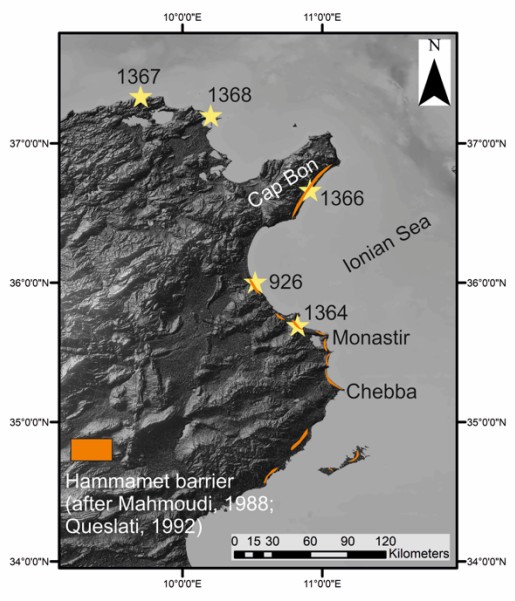

Fig. 5. The westernmost Ionian Sea coast (Tunisia). The datapoints included in the WALIS database are depicted with their
WALIS ID. Gebco data were obtained from GEBCO Compilation Group (2020) GEBCO 2020 Grid (doi:10.5285/a29c5465-
b138-234d-e053-6c86abc040b9). SRTM data were obtained from http://srtm.csi.cgiar.org (Jarvis et al., 2008).

4.4 Coastal lowlands

*Rosh Hanikra* (ID 928) and *Shavey Zion* (ID 935; Fig. 2) – in both sites the *Strombus bubonius*-bearing deposit is a poorly
sorted bioclast bearing conglomerate resting on the surface of an abrasion platform (marine terrace); depositional





environment is the swash zone of the beach (Sivan et al., 2016). The deposit is situated at 1.3 – 2.4 m (Sivan et al., 2016).

This is a facies-based sea-level indicator with an indicative range that includes storm wave swash height and breaking depth. The palaeo-sea level is at 1.8±1.0 m (average of *n*=3 indicators and standard deviation) during MIS 5e.

*Gulf of Gabès* (ID 1365; Fig. 4) – The LIG deposit is part of a coastline-parallel beach ridge. The lower part of the ridge is characterised by planar laminated beds of bioclastic sand bearing Strombus bubonius fossil remains (Gzam et al., 2016). The deposit is situated at 3 m (Gzam et al., 2016). This is a facies-based sea-level indicator with an indicative range of +3 m to 0

m including 1.5 m mean tidal range. The palaeo-sea level is at 1.5±3.4 m during MIS 5e.

4.5 Alpine orogenic belt

*Korba* (ID 1366; Fig 5) – This site is part of the Cap Bon barrier stretching almost parallel to the modern coastline (Elmejdoub and Jedoui 2009). The barrier shows siliciclastic sand in cross-bed or foreset bedding bearing Strombus bubonius in places (Elmejdoub and Jedoui 2009). The depositional environment is foreshore to beach (Elmejdoub and

Jedoui, 2009; Mauz et al., 2012). The elevation should be 5-10 m but is poorly defined. This is a facies-based sea-level indicator with an indicative range of +1 m to -3 m. The palaeo-sea level is at 13±10 m during MIS 5e.

*Rass Zebib* (ID 1368; Fig. 5) – the indicator is part of a cliff section the lower part of which is characterised by planar laminated beds of bio- and siliciclastic sand of shoreface to foreshore environment (Mauz et al., 2009). The elevation should be around 5 m but is poorly defined. This is a facies-based sea-level indicator with an indicative range of -3 m to -8 m. The

palaeo-sea level is at 11±6 m at 131±7 ka (Mauz et al., 2009).

*Ras el Korane* (ID 1367; Fig. 5) - the indicator is part of a cliff section the middle part of which is characterised by low-angle cross bedded bioclastic calcarenites (Wided et al., 2019). The depositional environment is foreshore. The elevation should be 5-6 m but is poorly defined. This is a facies-based sea-level indicator with an indicative range of -1 m to -5 m. The palaeo-sea level is at 8±5 m during MIS 5e as deduced from facies correlation (Wided et al., 2019).




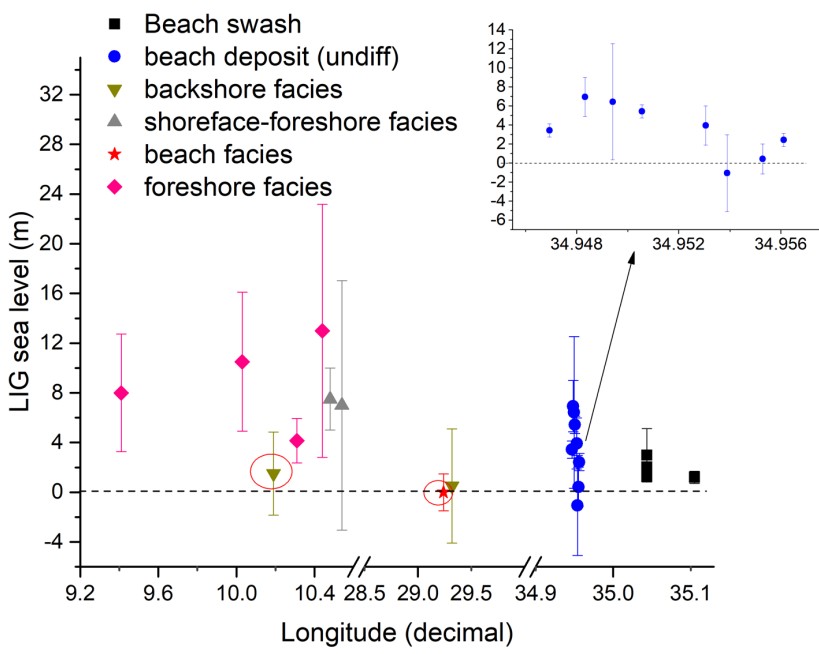

Fig. 6. The eastern Mediterranean sea-level indicators plotted against their longitudinal position. Red circle highlights datapoints potentially affected by non-GIA processes. The inset displays the sites on Carmel coast (Israel; Galili et al., 2007).

## 5 Further details

### 5.1 Impact of GIA mechanisms on LIG shoreline elevation


Similar to other mid-latitudinal coastal areas (e.g., US-Carolina coast; Parham et al., 2007) controversies exist on the position of the MIS 5 shorelines. In fact, in none of the tectonically dormant coastal zones reviewed here the LIG shoreline is observed at the commonly expected 6 m altitude (Fig. 6). Furthermore, for the MIS 5a Dorale et al. (2010) indicate that the shoreline in the West Mediterranean should have been close to modern MSL, but at Hergla (ID 926) the shoreline is at ~4m


(Mauz et al., 2018). Following results from geophysical modelling (e.g., Creveling et al., 2015) even the far-field shoreline can depart by up to 4 m from the eustatic value owing to local dynamic topography (e.g., Austermann et al., 2017). On the other hand, the LIG eustatic level is reconstructed probabilistically with a 4 m uncertainty (Kopp et al., 2013). Transferring these findings to the eastern Mediterranean, we can say that on tectonically dormant coasts the LIG sea level must have been situated at 5 ± 8 m. In fact, most of our datapoints depicted in Fig. 6 fall in this range of -3 m to 13 m.




## 5.2 Last Interglacial sea-level fluctuations

Sea-level fluctuation within LIG is reported from two sites: one site is Rosh Hanikra (ID 928) where the sea level fell from 3.3 m to 0.8 m and rose again to 6.5 m (Sivan et al., 2016). The other site is Hergla (ID 926) where the sea level should have risen first to 3 m, then dropped to near or below MSL and rose again to 4 m (Hearty et al. 2007). For this latter site, however,

Mauz et al (2018) showed that the upper marine deposit at 4 m is younger than LIG.

## 6 Future research directions

With regard to LIG sea-level research the most interesting sites are situated on the African passive continental margin where tectonic movements are minimal and sediment supply is sufficient to generate a datable sea-level indicator with a small vertical uncertainty. Coastal zones that merit further attention are (1) the Carmel coast (Israel), where the LIG highstand

formed a sediment wedge in onlap architecture (Fig. 2), and (2) the north Africa coast where LIG sediments formed a large barrier preserved in patches (Figs 4 and 5). In addition, the datapoints depicted in Fig. 6 that do not fall in the predicted range of 5±8 m LIG sea-level elevation deserve further analysis.

## 7 Data availability

The two databases are available open access and are kept updated as necessary at the following links:

http://doi.org/10.5281/zenodo.4274178 (Sivan and Galili, 2020) and http://doi.org/10.5281/zenodo.4283819 (Mauz, 2020). The files at these links were exported from the WALIS database interface on 15 November 2020 and on 21 November 2020, respectively. Description of each data field in the database is contained at this link: https://doi.org/10.5281/zenodo.3961543 (Rovere et al., 2020), that is readily accessible and searchable here: https://walishelp.readthedocs.io/en/latest/. More information on the World Atlas of Last Interglacial Shorelines can be found here: https://warmcoasts.eu/world-atlas.html. Users of our database are encouraged to cite the original sources in

addition to our database and this article.

### 8 Author contribution

B.M. was primary author and responsible for writing of the text and design of the figures, including all entries into WALIS regarding the eastern Mediterranean (except Israel). D.S. and E.G. compiled the data for the Israel coast and are responsible for relevant entries into WALIS.

### 9 Competing interests

The authors declare that they have no conflict of interest.

### 10 Acknowledgements

We thank Lotem Robins (Haifa) for uploading the Israel data to the WALIS database, Sara Stuecker (Salzburg) for generating the DEMs used for the map-based figures and Noureddine Elmejdoub (Gabès) for compiling the Tunisia data.

The data described in this paper were compiled in WALIS, a sea-level database interface developed by the ERC Starting Grant "WARMCOASTS" (ERC-StG-802414), in collaboration with PALSEA (PAGES / INQUA) working group. The database structure was



designed by A. Rovere, D. Ryan, T. Lorscheid, A. Dutton, P. Chutcharavan, D. Brill, N. Jankowski, D. Mueller, M. Bartz, E. Gowan and K. Cohen.

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
