# Peer review of "MIS 5e sea-level proxies in the eastern Mediterranean coastal region"

_Earth System Science Data, 2020_

## Referee Comment (RC1) · Anonymous Referee #1 · 30 Dec 2020

I read with interest the manuscript by Mauz, Sivan and Galili, MIS 5e sea-level proxies in the eastern Mediterranean coastal region that was submitted in the framework of special issue of ESSD. Authors present an assessment of the eastern Mediterranean evidence of last interglacial shorelines according to the international protocol for sea-level studies. They reported 21 records of MIS 5e along the Israeli, Egypt and Tunisian coasts. The text is easy to read and it fits well the scope of the special issue. However, I do not think authors performed the review of last interglacial shorelines in the eastern Mediterranean, as they stated in title. A large part of the eastern Mediterranean data is not included in the database (eg. Greece, Cyprus and much of the Turkish coast). So, the title does not reflect the content of the paper. I understood that authors preferred to focus on the passive margins which are less affected by tectonic influence but, in

my opinion, it is very important to have a clear picture of the whole basin. This is important to disentangle the different factors influencing the current elevation of LIG shorelines. As an example, these data can be useful to make comparison with more recent uplift/subsidence trends (at Holocene and decadal scale). Furthermore, authors focused a large portion of the text (section 1.1.1) on the active coastal zones. This part is floating in text because authors never approached the tectonic control on LIG shorelines in the remaining part of the manuscript. Moreover, in Tunisia there is a large amount of literature in French that is never mentioned in text (e.g., Mahmoudi 1986; 1987; 1988; Sorel and Kamoun, 1980; Paskoff and Salanville, 1980; 1983, 1986 etc). I'm sorry to say that authors cannot ignore this literature. This is the complex part of compiling a database that must be comprehensive and not just ignore the non-english literature published on peer reviewed journals. So, in my opinion authors should include all the data of the eastern Mediterranean. If they decline to do this effort, they should change the title. Of course, the first option would be much important for the scientific community.

Definition of the Indicative meaning This is another weakness of the manuscript. In table 1 authors report 5 different typologies of sea-level indicators. First of all, it is unclear what is considered sea-level index point and what is considered marine limiting point. The nomenclature as well does not follow the international protocols (eg, Shennan et al., 2015). As an example, the coastal notch. The IR is MHW to MLW while the RWL (e.g., the midpoint of the IR) is (MHW to MLW)/2 Also the other types of RSL indicators are not well explained. They should be standardized and a clear explanation of each indicator should be given. I know that, for high energy coast, the IR is dependent by the local hydrodynamics but I do think this issue was well addressed in the recent protocol provided by Rovere et al., 2016. Furthermore, it is unclear why you grouped in a single typology all the sedimentary facies. They are very different (eg., a lagoon is from a low energy environment while a carbonate sand can deposit in a very energetic environment). The section 2.3 is not exhaustive, because the reported IR (1 to 4 m for foreshore and 4-8 for shoreface) were only seldom applied on the data.

[Figure]

Why, for instance authors selected +1 to -3 m as IR for Ras Karboub? I am not saying that this is incorrect but I think authors should better define these IR in the methods. Similarly, why authors selected 1 to 3 m of depth for the El Max Abu Sir? This is also not explained in the methods. I accessed the database and this detailed description is available in the RSL indicator tab. I suggest to transfer this part also in the main text. Another point related to the database in Tunisia. Why authors did not report the large number of U-series available in Jedoui et al., 2003 on QSR. I could not find this in the database. Is there a reason to exclude these data? If so, authors should explain this because the database must be as much inclusive as possible.

I have also noted some mismatches in the Israeli dataset. In particular, the data from the Carmel coast from Galili et al 2007 are not supported by stratigraphic information. Authors transformed them in sea level indicator and not in marine limiting points. Authors should justify this choice because this is not in line with the methods of the manuscript and the guidelines of Walis. I went inside the database and I noticed, for instance, that a vermitid (933) is listed as a sea-level indicator in the database but in section 2.5 this is reported as marine limiting point. I found many other inconsistencies between text and database.

This confusion in the data is also present in the Figure 6. Here authors report a number of data which not reflect the earlier part of the paper. The indicative range of each of the sea-level data listed in the figure must be well explained in the methods. Why the authors did not show the marine limiting points on this figure? They can be very important to understand the general pattern. All these issues must be solved in order to make this paper useful for future research, as requested by the scope of the journal and of the special issue. In its present form, the paper is weak and incomplete. So, I suggest a major revision before the publication. This new version should clearly state why this database is not showing all the data of the eastern Mediterranean (please change the title) and should be significantly implemented in its methodological part. Authors must clearly define what was considered a marine limiting point and what was consid-

ered a sea-level indicator. For the latter, authors should detail the different IR used for each of them. Author should carefully cross-check the information they inserted in the database with those discussed in text. Finally, authors should discuss and report ALL the available literature and not just the one published in English journals.

———————————————

---

## Short Comment (SC1) · 31 Dec 2020

Thank you for the detailed review. I will look at all points of critique and respond accordingly.

———————————————————

---

## Referee Comment (RC2) · Anonymous Referee #2 · 1 Jan 2021

General comment

The paper by Mauz et al. provides an overview of sites in the eastern Mediterranean coasts where evidence of indicators of the Last Interglacial shoreline were previously published. The compilation was taken in the frame of the WALIS project - the World Atlas of Last Interglacial Shorelines and follows its protocol as supporting databases show. To my knowledge a modern compilation of LIG data from the eastern Mediterranean is lacking so the effort is welcomed.

However, I found the compilation necessary but not exhaustive. I do not understand why, and the authors should explain it, much of the coasts on the northern side where regional tectonics strongly affect the present position of the LIG shoreline (Greece, Turkey), are excluded from the database. The authors state in section 4 that they

designedly exclude those sites affected by non-GIA processes, and this would be a good idea in case they want to pursue a research task, and specifically to compare their elevation data to GIA predictions to test model scenarios or look for minimal coastal displacements unaccounted for by these models. This is obviously not the case here because they do include some sites in regions clearly experiencing active deformation such as northern Tunisia, the Marmara Sea and the Carmel coast. The fact that for a number of reasons, including low displacement rates or geometric characteristics of the active faults in the selected areas (most faults are strike-slip or thrusts), the LIG shoreline does appear close to the eustatic position - unlike what happens for instance in the Corinth Gulf - is not a justification and actually could lead to wrong estimation of GIA model parameters.

On the same reasoning, I found confusing that they organize the literature overview description by following geodynamic provinces and do a great effort in describing coastal zones affected by active tectonics and with reported evidence of LIG indicators. This part is not utilized in their compilation so there is an apparent discrepancy between text and supplemental material. In my opinion, overview papers such this, which is related to an Atlas, should encompass all available data and not just a selection of them. The Authors should include much more published data and discuss what processes control the elevation of LIG shoreline in different sectors to make this paper more appealing to the community.

In the databases, I found some conflicting definitions and typing. For example, the sea-level indicator that in one DB is defined "Beach swash deposit" is in some cases undistinguishable in terms of lithofacies from what in the other DB is labelled as "Foreshore" or "Coastal Barrier". In some entries, there is confusion between chronostratigraphy and lithofacies description (e. g. Indicator description of row 11 in Israel database). In general, the database tables are hard to follow because of the large number of entries. I think this is a WALIS template problem and not specific to this paper, and I understand that authors may have problem in homogenising. However, tables in the

text paper should be more concise and to the points, and I find tables 1 useful but too generic, and table 2 useless. Table 1 lists some indicators that are not found in the provided databases (e. g. marine terraces, Cladocora reef, Lithophaga holes). I acknowledge that terraces are found mostly in the tectonic unstable zones, which are left out of the database; but then, why do you quote it in the paper? The sediment facies indicator, which is the most largely used one, is rather condensed in text table. In the text there is a description of bathymetric corrections, but I could not understand from the xsl tables whether they were applied or not (I understand they were not).

A further weak point regards organization of description of zones. In section 1.1 (overview) it is stated that the description follows division of zones relative to tectonic structures, but I have a difficulty in following the adopted criteria:

1.1.1. Active tectonic zones is based on tectonics

1.1.2 Nile littoral cell and delta is based on oceanography and morphology.

1.1.3 African passive margin is based on tectonics, but the Nile is part of it.

1.1.4 Coastal lowlands is based on morphology...could be either passive or active margins. The ones you identify are part of the African passive margin

In section 4 (E Med RSL sites) you introduce zones different than the description in Literature Overview section. For instance, Black Sea rift and Alpine orogenic belt (which are active tectonic zones) Figure 6: It is quite difficult to relate this figure to the text and to the electronic database, in light of the lack of ID numbers in figure (checking it with the Longitude is impossible). Also, it is not easy to understand why error bars on same indicators (e. g. foreshore facies) are different. I presume they reflect the sum of uncertainties. A discussion on uncertainties is lacking in the paper (but this is a pitfall of the database as well, where the uncertainty estimation strategy is unclear).

Specific comments and technical corrections

Line 21: Specify in which reference frame Africa moves. The African plate is moving

generally NE in a hotspot reference frame, and is moving from NE to NW moving from E to W along its margin relative to Europe.

Line 43: Strombus bubonius (LMK) is today identified as Strombus (=Persististrombus) latus GMELIN

Line 123: Mauz et al 2009 is not in reference list

Line 126: Black Sea is included in Coastal lowlands but it is not shown like this in Fig. 1

Line 132: The Ahihud fault separates the Rosh Hanikra platform from Haifa bay. Here and elsewhere, these local features are distracting the reader as long as they do not impact the position of the LIG shoreline, or they do it but they are not shown on a map.

Line 137: How do you know it is LIG shoreline?

Line 139: Gharbi et al is not in reference list

Line 142: LIG deposit is part of a beach ridge stretching parallel to the modern coastline at ca 3 m altitude. Add reference.

Line 171: where the amplitude is around 70 cm. at line 140 m you state the tide amplitude is 1.5 m

Line 184: where did you take these depths from? Add a reference

Line 210: Bosellini et al is not in reference list

Line 225: In the Isreael database there are 5 more indicators for a total of 26, not 21

Line 240: 2.3 mm/a subsidence is a pretty high estimate...with this velocity the LIG shoreline should be 230 m below sea level. Please clarify

Line 321: local dynamic topography. How do you know is dynamic topography only and not unaccounted GIA effects or compaction or some local tectonics?

Line 326: Please specify time scale of fluctuations.

Line 330: how much younger?

Line 331: Future research directions should be modified according to the suggests paper rearrangement.

---

## Short Comment (SC2) · 7 Jan 2021

Reply to Reviewer#1

I am sorry to say that the authors could not achieve agreement on how to reply to the review. The text below is therefore from me only. A separate comment from the co-authors may follow.

I thank the referee for their expert review and wish to reply as follows.

*"I do not think authors performed the review of last interglacial shorelines in the eastern Mediterranean, as they stated in title A large part of the eastern Mediterranean data is not included in the database (eg. Greece, Cyprus and much of the Turkish coast)."*

The title is about "MIS 5e sea-level proxies" which is not quite the same as "last interglacial shorelines". The active tectonic zone of Greece, Turkey, etc was included in the literature overview, hence the title "eastern Mediterranean" and later excluded from the database for reasons outlined in the introduction ("…*enables us to separate shoreline data generated to unravel tectonic processes from sea-level data generated to reconstruct the LIG sea level and the associated ice volumes, eustacy and related GIA processes"*). By separating the data I have been following two of the three aims of the WALIS project: first, address the uncertainties associated with the 6-10 m LIG highstand, and second, understand LIG sea-level oscillations. By cleaning the data from non-GIA components we followed the key objective of the project: analysing the database in the light of a large array of GIA models. See https://cordis.europa.eu/project/id/802414/reporting for details.

*"it is very important to have a clear picture of the whole basin. This is important to disentangle the different factors influencing the current elevation of LIG shorelines."*

I agree and this is exactly what the manuscript intended to deliver: differentiate between GIA and non-GIA affected coastal zones.

*"Authors focused a large portion of the text (section 1.1.1) on the active coastal zones. This part is floating in text because authors never approached the tectonic control on LIG shorelines in the remaining part of the manuscript"*

I am not sure if I fully understand this comment. What I can say is that the paper guideline provided by Alessio says about sec 1 (Literature overview): "*Give a brief overview of the historical development of MIS 5e sea level reports in the study area*." I feel this is what I did.

*"in Tunisia there is a large amount of literature in French that is never mentioned in text… authors cannot ignore this literature"*

A number of languages are spoken in the eastern Med (Greek, Turkish, Hebrew, Arabian…) and some papers on Quaternary geology are published in these languages. Also, a number of Europeans have been working in North Africa during the early to late 20[th] century and most of them published in their home language (French, Italian, German, Polish…). I am sorry to say that I am unable to read these languages and I believe this lack of ability applies to many colleagues. I cannot see why we should give preference to the French literature over, say, Turkish or Arabian papers. Because English is the scientific language, for the sake of reproducibility the review has to focus on literature published in English.

*"If they decline to do this effort, they should change the title. Of course, the first option would be much important for the scientific community."*

I appreciate the referee's opinion about what is important for the scientific community.

*"in table 1 it is unclear what is considered sea-level index point and what is considered marine limiting point."*

Thank you for highlighting this – the header of the table should say "RSL proxies reviewed in this study".

*The nomenclature as well does not follow the international protocols (eg, Shennan et al., 2015). As an example, the coastal notch. The IR is MHW to MLW while the RWL (e.g., the midpoint of the IR) is (MHW to MLW)/2."*

I am afraid, I do not find an IR definition for coastal notch in Shennan et al. 2015. In fact, we have used Antonioli et al 2015 as reference.

*Also the other types of RSL indicators are not well explained. They should be standardized and a clear explanation of each indicator should be given. I know that, for high energy coast, the IR is dependent by the local hydrodynamics but I do think this issue was well addressed in the recent protocol provided by Rovere et al., 2016*

I appreciate this comment – there is definitely a lack of standardisation in table 1. The referee recommends using local hydrodynamics as outlined in Rovere et al., 2016 where RWL of a beach deposit is half-way between wave breaking depth and berm. This is reasonable for the Holocene beach for which instrumental record is available. For earlier interglacials the berm (or similar landform) has to be reconstructed through sedimentary features and the breaking depth remains unknown as long as coastal topography is not perfectly reconstructed and wave climate is deduced from downscaled palaeo-climate models. For the purpose of the review and compilation of LIG sea-level data we need to find criteria beyond hydrodynamics.

*it is unclear why you grouped in a single typology all the sedimentary facies. They are very different (eg., a lagoon is from a low energy environment while a carbonate sand…"*

I do concur with the referee, single typology is indeed not sufficient to describe the IR of a coastal deposit. I will separate backshore, foreshore and shoreface deposits in the table and ascribe IRs to the deposits. For reasons outlined above the IRs will not be deduced from hydrodynamics but from bed forms and facies following descriptions in textbooks such as Reading (1986). Facies relationships are deduced from standardised principles of Walter's Law and Sequence stratigraphy as described by Nelson (2015) in the Handbook.

*The section 2.3 is not exhaustive, because the reported IR (1 to 4 m for foreshore and 4-8 for shoreface) were only seldom applied on the data.*

I am not sure if I understand this comment – I suspect that the reviewer is puzzled by some discrepancies between our facies description and the representation of the respective datapoint in the database. This is indeed a problem that we need to sort out together.

*authors should better define these IR in the methods*

A method section does not exist in the review paper. Table 1 summarises the IRs.

*Why, for instance authors selected +1 to -3 m as IR for Ras Karboub? I am not saying that this is incorrect but I think authors should better define these IR in the methods. Similarly, why authors selected 1 to 3 m of depth for the El Max Abu Sir? This is also not explained in the methods. I accessed the database and this detailed description is available in the RSL indicator tab. I suggest to transfer this part also in the main text.*

Table 1 summarises the IRs and list the references from which the IRs are deduced. I agree with the referee that there are discrepancies between our facies description and the RSL indicator description in the database. The problem is not solved by copying the database text into the review.

*Why authors did not report the large number of U-series available in Jedoui et al., 2003 on QSR?*

Jedoui et al dated the Jeffara coastal barrier using mollusc shells. This barrier is continuous from NW Libya to the Gulf of Gabes and then continuous in the Gulf of Hammamet up to Cap Bon. The barrier has been dated in several places by OSL technique which is, in this particular case, a better suited dating technique than U-series. Critical evaluation of analytical data (see U-activity ratio in Jedoui's Table 1) is a necessary consequence of known methodological problems (e.g., Kaufman et al. 1996; Geochem and Cosmochem Acta 60) and thus, why populating the database with data that do not survive state of the art screening?

*Authors transformed them [Israeli dataset] in sea level indicator and not in marine limiting points. Authors should justify this choice because this is not in line with the methods of the manuscript and the guidelines of Walis.*

See separate comment from co-authors.

---

## Short Comment (SC3) · 7 Jan 2021

Comment on Reviewer#2

I am sorry to say that the authors could not achieve agreement on how to reply to the review. The text below is therefore from me only. A separate comment from the co-authors may follow.

I thank the referee for their expert review and wish to reply as follows.

*I do not understand why, and the authors should explain it, much of the coasts on the northern side where regional tectonics strongly affect the present position of the LIG shoreline (Greece, Turkey), are excluded from the database*

The active tectonic zone of Greece, Turkey, etc was included in the literature overview, and later excluded from the database for reasons outlined in the introduction ("…*enables us to separate shoreline data generated to unravel tectonic processes from sea-level data generated to reconstruct the LIG sea level and the associated ice volumes, eustacy and related GIA processes*").

*"…designedly exclude those sites affected by non-GIA processes, and this would be a good idea in case they want to pursue a research task, and specifically to compare their elevation data to GIA predictions to test model scenarios"*

My reading of the WALIS project text is exactly this – pursue a research task by comparing GIA models with the database. The text says: "we are working with earth modelers towards obtaining a large array of models to predict vertical land motions caused by glacial isostatic adjustment and dynamic topography. Analyzing the database in light of these models will allow us to give a more precise answer to the question: "how high was sea level in the Last Interglacial?" See https://cordis.europa.eu/project/id/802414/reporting for details.

*This is obviously not the case here because they do include some sites in regions clearly experiencing active deformation such as northern Tunisia, the Marmara Sea and the Carmel coast.*

I share these sceptical thoughts. Some arguments convinced me to include the sites in the database:
northern Tunisia: the terrain is thrust folded. Quaternary terrestrial and marine deposit do not show deformation and overlie deformed Pliocene strata unconformably in graben, synclines and depressions. The LIG deposits are part of cliff sections, marine terraces are absent. Some data (e.g. striated faults, flexure, seismic; Essid et al., 2016, Tectonophysics 682; Melki et al 2011, J Geodyn. 52) suggest ongoing compression (not uplift!) but whether LIG coastal deposits are affected is not clear.
Marmara Sea: I guess the referee means ID 927 which is described in sec 4.1 (Black Sea). The site was included because tectonic land movements are quantified (Avsar et al 2017) and because its elevation is within the 5±8 m as expected for the LIG (see sec 5.1 for details).
Carmel coast: looking at published data (e.g. Gvirtzman et al 1997; Porat et al, 2003) a SSW-directed tilt of the shelf seems obvious. Mauz et al 2013 show that around 20 km south of the Walis ID942 site the dip of the LIG deposit changes from gentle (1-4 degree) to steep (~10 degree) suggesting that the northernmost part of the Carmel coast is unaffected by the tilt.

*The fact that for a number of reasons, including low displacement rates or geometric characteristics of the active faults in the selected areas (most faults are strike-slip or thrusts), the LIG shoreline does appear close to the eustatic position - unlike what happens for instance in the Corinth Gulf - is not a justification and actually could lead to wrong estimation of GIA model parameters.*

I appreciate this comment, the reviewer makes a good point. If proxy data are included in the database just because the indicator is situated close to an inferred eustatic position, the GIA modeler could perhaps adjust the melting history accordingly. However, this type of approach is prone to circular conclusion and serious modelling would not proceed in this way. Nevertheless, we should try and facilitate modelling work by include critical evaluation of the proxy data in our data compilation work. This includes, for instance, mentioning faults, even when their activity is unknown for the period of interest.

*In my opinion, overview papers such this, which is related to an Atlas, should encompass all available data and not just a selection of them. The Authors should include much more published data and discuss what processes control the elevation of LIG shoreline in different sectors to make this paper more appealing to the community.*

In the second paragraph of this review the referee asked us to explain the data selection, here however inclusion of "much more data" is requested. I am therefore feeling a little unsettled about this comment. In addition, "*discussion of processes that control elevation*" is requested. I understand that the ESSD journal does not envisage discussion of data. In fact, the paper template does not include a "Discussion of data" section. This is what Alessio emailed the authors in September 2020:

"*Scope of the paper. After communications with the Chief Editor of ESSD, David Carlson, we established that every paper should be framed as the description of a data product. The data product is the regional database you are assembling through our system. … The MS should contain short overviews of the sites inserted in the database, also detailing the choices you make in including / excluding sites, recalculating uncertainties, assessing quality.*"

*I acknowledge that terraces are found mostly in the tectonic unstable zones, which are left out of the database; but then, why do you quote it in the paper?*

On Galilei coast LIG deposits are found on a "abrasion platform" (Sivan et al., 2016). For further details see comment of co-authors.

*I have a difficulty in following the adopted criteria [regards organization of description of zones].*

I understand, it is indeed not consistent. The zones were classified according to the dominant controlling factor. The Levant coast is definitely governed by the Nile cell and by the sediment load of the delta; the gulfs and embayments are best described as coastal lowlands because 'Rift zone" (Gulf of Gabes, Gulf of Sirte) or "Graben" (Haifa Bay) would allude to tectonic processes which are not confirmed to be active in the late Quaternary. I wouldn't know how to better classify the coastal zones other than not to classify at all.

*In section 4 (E Med RSL sites) you introduce zones different than the description in Literature Overview section*

Thank you for highlighting this inconsistency – it applies to Black Sea and north Tunisia. It will be ironed out.

*Figure 6: It is quite difficult to relate this figure to the text and to the electronic database, in light of the lack of ID numbers in figure*
Thanks for highlighting the weakness – The ID numbers are now added in the figure and the caption will be changed accordingly.

[Figure]

*it is not easy to understand why error bars on same indicators (e. g. foreshore facies) are different. I presume they reflect the sum of uncertainties. A discussion on uncertainties is lacking in the paper (but this is a pitfall of the database as well, where the uncertainty estimation strategy is unclear).*
Uncertainties are key to the WALIS aims and there is a lot to say about error estimation. As an author I can say that the error bars displayed in Fig. 6 are obtained from the Walis database software. The user has no influence on the calculation. The online platform does however outline the equation used for uncertainty calculation (simple quadratic square root rule) and using this simple equation I obtained identical uncertainty values up to one digit. And yes, the error bar represents the sum of all uncertainties. To assess the standardisation of indicators the IR error should be examined. The difference between IR-errors of the same indicator should reflect the tidal amplitude and if not, the standardisation is imperfect.

*Specific comments and technical corrections*
Thanks for these – much appreciated

*Strombus bubonius (LMK) is today identified as Strombus (=Persististrombus) latus GMELIN* – what is today and what is the reference?

*The Ahihud fault separates the Rosh Hanikra platform from Haifa bay. Here and elsewhere, these local features are distracting the reader as long as they do not impact the position of the LIG shoreline, or they do it but they are not shown on a map* – I believe the faults have to be mentioned because there is no LIG shoreline onshore in Haifa bay while the shoreline is evident adjacent to the bay at 1-2 m on Galilei coast and at 0-7 m on Carmel coast. Not mentioning the faults would look as

if the authors of the review haven't noticed the problem. For more details see comment of the co-authors.

*Line 137: How do you know it is LIG shoreline?* Giglia (1984) describes a "shallow marine-beach deposit", the stratigraphic context indicates "Tyrrhenian transgression". His map shows the distribution of the deposit in the coastal plain of the Sirte Gulf.

*Line 171: where the amplitude is around 70 cm. at line 140 m you state the tide amplitude is 1.5 m* – in line 171 the text is about tidal amplitude, in line 140 the text is about tidal range.

*Line 184: where did you take these depths from? Add a reference* – the references are in table 1.

*Line 240: 2.3 mm/a subsidence is a pretty high estimate...with this velocity the LIG shoreline should be 230 m below sea level. Please clarify.* – thanks for pointing this out. I shall make clear in the text that the estimate is for the most recent period (2008-2014) of instrumental record.

*Line 321: local dynamic topography. How do you know is dynamic topography only and not unaccounted GIA effects or compaction or some local tectonics* – this comment is unclear to me. The text in line 321 summarises the results from Austermann et al.'s modelling work.

*Line 326: Please specify time scale of fluctuations* – the Walis project is about LIG and the Walis texts says: "Last Interglacial (here intended as MIS 5e, peaking 125 thousand years ago)".

*Line 330: how much younger?* I don't think this question relates to the manuscript under review. Please see the reference for details.

*Line 331: Future research directions should be modified according to the suggests paper rearrangement*. For this section Alessio's guideline says: "What is needed to improve the MIS 5e record in the area studied? Are there "hotspot sites"? I feel this is exactly what we did.

---

## Short Comment (SC4) · 12 Jan 2021

12.1.2021

To:

The editors of the Special Issue: WALIS - the World Atlas of Last Interglacial Shorelines
Following the reviewers comments accepted for the manuscript titled "MIS 5e sea-level proxies in the eastern Mediterranean coastal region" by Barbara Mauz et al.
MS No.: essd-2020-357

We would like to thank the two reviewers for the detailed and thorough reviews they performed. We agree with almost all of their remarks, and we believe that following their recommendations will considerably improve the article.

From the very beginning we (all authors) decided to share the responsibility; Sivan and Galili- the data from Israel and Cyprus while the data from north Africa, Greece and Turkey- Dr Mauz.

During the writing process we respected the decisions of the first author, although in many cases we disagree with them. We wrote the needed chapters related to Israel and Cyprus as agreed.

Following the comments the reviews made, we clarified to Dr Mauz that most of the comments are reasonable and that following them will considerably improve our paper. In our response to the "letter to the editor" draft sent to us by Dr. Mauz we have been trying to respect the first author decisions on one hand, and to follow most of the important recommendations of the reviewers, on the other hand. While doing this, Dr. Mauz already published a note stating that there is disagreement between the co-authors on the corrections needed.

Therefore, below you will find our brief point-by-point answers to the reviewers' comments. This is, to the best of our understanding, the way in which the reviewers' recommendations should be carried out, and we call Dr Mauz to follow them. The most significant issues required are: adding and citing the available, missing works and data from the studied areas, mainly from Cyprus and the works done in North Africa.

We are looking forward to see the final version of the article after the corrections, before it is submitted to the journal.

Please see our detailed answers to the reviewer's comments (the comments are in red while our answers are in black):

**Comment on Review#1**

"I do not think authors performed the review of last interglacial shorelines in the eastern Mediterranean, as they stated in title. A large part of the eastern Mediterranean data is not included in the database (eg. Greece, Cyprus and much of the Turkish coast)."
We agree. We insisted on adding at least the data from Cyprus, an Island that contains large amount of MIS 5.5 data, large part of it with high quality elevation measurements and good quality of dating. We insisted on adding this data but the corresponding author decided not to include it.

"it is very important to have a clear picture of the whole basin. This is important to disentangle the different factors influencing the current elevation of LIG shorelines."

We agree and this is exactly what the manuscript intended to deliver: to present all data from various present-day elevations. A database that will allow further research to study the various mechanisms involved.

"Authors focused a large portion of the text (section 1.1.1) on the active coastal zones. This part is floating in text because authors never approached the tectonic control on LIG shorelines in the remaining part of the manuscript"As mentioned above, we believe that all data has to be presented in such a review paper. The tectonic and/or GIA contribution can be discussed in short but in general this kind of paper has to present all data available and the uncertainties it contains.

"in Tunisia there is a large amount of literature in French that is never mentioned in text… authors cannot ignore this literature"
We agree. The issue of using at least French literature was discussed few times but Dr. Mauz didn't accept our approach. The south and east Mediterranean was historically divided between French speaking authorities like most of north Africa, Syria and Lebanon, while other areas like Palestine (now Israel) and Cyprus were under English speaking authority. In Turkey there is a lot of research published in German. A review paper has to include all kind of reported scientific data. Unfortunately, the corresponding author didn't share with us the same understanding.

"If they decline to do this effort, they should change the title. Of course, the first option would be much important for the scientific community."
We prefer to add more data but we also agree to the less preferable option of changing the title.

"in table 1 it is unclear what is considered sea-level index point and what is considered marine limiting point."
This recommendation was accepted and the table was revised.

The nomenclature as well does not follow the international protocols (eg, Shennan et al., 2015). As an example, the coastal notch. The IR is MHW to MLW while the RWL (e.g., the midpoint of the IR) is (MHW to MLW)/2."
The corresponding author answered that "we have used Antonioli et al 2015 as reference" and we accept it.

Also the other types of RSL indicators are not well explained. They should be standardized and a clear explanation of each indicator should be given. I know that, for high energy coast, the IR is dependent by the local hydrodynamics but I do think this issue was well addressed in the recent protocol provided by Rovere et al., 2016
We accept the detailed answer of the corresponding author.

It is unclear why you grouped in a single typology all the sedimentary facies. They are very different (eg., a lagoon is from a low energy environment while a carbonate sand…"
Here too we accept the detailed answer made by the corresponding author.

The section 2.3 is not exhaustive, because the reported IR (1 to 4 m for foreshore and 4-8 for shoreface) were only seldom applied on the data
Please see the corresponding author response.

authors should better define these IR in the methods

Please see Table 1.

Why, for instance authors selected +1 to -3 m as IR for Ras Karboub? I am not saying that this is incorrect but I think authors should better define these IR in the methods. Similarly, why authors selected 1 to 3 m of depth for the El Max Abu Sir? This is also not explained in the methods. I accessed the database and this detailed description is available in the RSL indicator tab. I suggest to transfer this part also in the main text.

Please see the answer by B. Mauz.

Why authors did not report the large number of U-series available in Jedoui et al., 2003 on QSR?

The sites mentioned in the areas that Dr. Mauz reported on and she is the expert in OSL and therefore we account on her response.

Authors transformed them [Israeli dataset] in sea level indicator and not in marine limiting points. Authors should justify this choice because this is not in line with the methods of the manuscript and the guidelines of Walis.

The Israeli data is not well presented in the paper. We had a lot of disagreements with the corresponding author. This bad representation leads to misunderstandings: in the Galilee type section of Rosh Hanikra (Sivan et al., 2016) we have at least three different kinds of RSL indicators: we have the Strombus bearing unit, the Vermetids that is to our understanding an index point and we have the bioclastic sandstone. This description has to be be added in order to show which part of the sequence is marine limiting (the bioclastic sandstone) and what are the index point (the Strombus and the vermetides units).

**Comment on Review#2**

I do not understand why, and the authors should explain it, much of the coasts on the northern side where regional tectonics strongly affect the present position of the LIG shoreline (Greece, Turkey), are excluded from the database.

This question was also addressed by the first reviewer. We agree and as we already mentioned above we think that the active tectonic zone of Greece, Turkey, and the coasts of Cyprus has to be included.

"…designedly exclude those sites affected by non-GIA processes, and this would be a good idea in case they want to pursue a research task and specifically to compare their elevation data to GIA predictions to test model scenarios"

To our opinion all sites from all kinds of coasts have to be presented so they can be used for all kinds of studies checking for tectonic and/or GIA rates and more.

This is obviously not the case here because they do include some sites in regions clearly experiencing active deformation such as northern Tunisia, the Marmara Sea and the Carmel coast.

The Carmel coast is part of the whole Israeli coast that is **relatively stable** (Galili et al., 2007; Sivan et al., 2016 and references therein).

The fact that for a number of reasons, including low displacement rates or geometric characteristics of the active faults in the selected areas (most faults are strike-slip or thrusts), the LIG shoreline does

appear close to the eustatic position - unlike what happens for instance in the Corinth Gulf - is not a justification and actually could lead to wrong estimation of GIA model parameters.

We have to present the data and later we can suggest mechanisms for various elevations. Only in a few sites GIA model predictions for the site at LIG were carried out. One of them is in Rosh Hanikra Israel (Sivan et al., 2016) where few models were checked. The results give an envelope but are not decisive.

In my opinion, overview papers such this, which is related to an Atlas, should encompass all available data and not just a selection of them. The Authors should include much more published data and discuss what processes control the elevation of LIG shoreline in different sectors to make this paper more appealing to the community.

Agree. See our answers above.

I acknowledge that terraces are found mostly in the tectonic unstable zones, which are left out of the database; but then, why do you quote it in the paper?

Cyprus has elevated terraces with well dated LIG deposits but unfortunately it was not included. It has to be in the paper.

I have a difficulty in following the adopted criteria [regards organization of description of zones].

It has to be changes.

In section 4 (E Med RSL sites) you introduce zones different than the description in Literature Overview section

It has to be corrected.

Figure 6: It is quite difficult to relate this figure to the text and to the electronic database, in light of the lack of ID numbers in figure

It was improved by Dr. Mauz.

[Figure]

it is not easy to understand why error bars on same indicators (e. g. foreshore facies) are different. I presume they reflect the sum of uncertainties. A discussion on uncertainties is lacking in the paper (but this is a pitfall of the database as well, where the uncertainty estimation strategy is unclear).

The co-author used the calculations made by the online WALIS platform.

**Specific comments and technical corrections**

*Strombus bubonius (LMK)* is today identified as *Strombus (=Persististrombus) latus GMELIN* – what is today and what is the reference?

We accepted the old terminology based on Sivan et al., (2016) about the Galilee coast, Israel where it is written: "In the Mediterranean Sea, the key fossil indicator for MIS5e is the gastropod Strombus bubonius. It is now known as Persististrombus latus (Taviani, 2014), but we retain the old synonym for ease of comparison with referenced literature (e.g., Zazo et al., 2003, 2013 and references therein; Bardaji et al., 2009, Sivan et al., 1999)".

In Cyprus, Galili et al. (2015) wrote that previous names of this index fossil were Strombus bubonius (Gignoux 1913; Deperet 1918) and Lentigo latus, and, more recently, Kronenberg & Lee (2007) suggested the name Persististrombus latus. Since the term 'Strombus bubonius' appears to be very common in the literature, including articles quoted in here, we abbreviate its name as 'SB' throughout this work

The Ahihud fault separates the Rosh Hanikra platform from Haifa bay. Here and elsewhere, these local features are distracting the reader as long as they do not impact the position of the LIG shoreline, or they do it but they are not shown on a map

We didn't want to present these faults since they are misleading but the corresponding author did add them.  The faults do not have any impact on the LIG RSL indicators. The paper deals with indications for sea levels not with coastlines. Therefore, the most important issue is the fact that the data from the Galilee and from the Carmel coast presents same LIG sea levels, in the frame of the uncertainties.

Line 137: How do you know it is LIG shoreline?

This question got an answer by Dr. Mauz.

Line 171: where the amplitude is around 70 cm. at line 140 m you state the tide amplitude is 1.5 m – Again, there is an answer by Dr. Mauz who was on charge of these sites.

Line 184: where did you take these depths from? Add a reference

Dr. Mauz already answered: the references are in table 1.

Line 240: 2.3 mm/a subsidence is a pretty high estimate...with this velocity the LIG shoreline should be 230 m below sea level. Please clarify.

Dr. Maus wrote that she'll make it clearer  since  the estimate is for the most recent period (2008-2014) of instrumental record

Line 321: local dynamic topography. How do you know is dynamic topography only and not unaccounted GIA effects or compaction or some local tectonics

Dr. Maus already answered that the text in line 321 summarises the results from Austermann et al.'s modelling work.

Line 326: Please specify time scale of fluctuations

In most cases there is no way to present time scale fluctuations. Even in Rosh Hanikra, Galilee coast, fluctuations were found based on field relations but with no way to date each one of them (Sivan et al., 2016).

Line 330: how much younger?

There is no way to know.

Line 331: Future research directions should be modified according to the suggests paper rearrangement.
Agree. It will be modified by the re-arrangement.

Sincerely,
Prof. Dorit Sivan                                                        Dr. Ehud Galili

E. GALILI

Dorit Sivan.

Copy: Dr Barbara Mauz-corresponding author

---

## Short Comment (SC5) · 14 Jan 2021

I find the comment posted by Dorit Sivan and Ehud Galili deeply disturbing. In my eyes it includes perverted facts and false accusations. For example the Cyprus database: we agreed that DS and EG would work on the data from Cyprus but they have failed to compile the data and to produce the database. Still, DS and EG accuse me: "We insisted on adding this data but the corresponding author decided not to include it." Other examples include apparent misrepresentation of Israeli data in the text, the inclusion of marine terrace as indicator in table 1 or the tectonic stability of the Carmel coast. I feel the way some of the comments are expressed cannot be regarded as scientific exchange and cast doubt on the integrity and honesty of DS and EG. I am deeply concerned that the discussion platform, provided by the publisher for the purpose of scientific exchange, is used as a vehicle to discredit me in public. I do now call on the editor to comment on this before the open discussion closes, to retract the comment of DS and EG and ask them to delete all personal accusations, return to scientific discussion and follow Copernicus best practice.

––––––––––––––––––––––––––––––

---

## Editor Comment (EC1) · Alessio Rovere (Editor) · 18 Jan 2021

Dear Authors,

I now had time to read the reviews of your paper and your successive comments. First of all, I would like to commend the work of the two reviewers. In my opinion, they approached the review with a constructive approach, raised valid concerns, and suggested several ways to improve the quality of your work. From your answers, it is clear that there is a fundamental disagreement in how you wish to respond to these comments, and I find it very unfortunate that you did not follow common scientific practice and decided not to discuss these matters internally and come to a mutual agreement on how to proceed with the paper revisions.

[Figure]

I was advised by journal managers that the journal cannot remove comments that are made public. If your disagreement is final, the reasonable thing to do at this stage from an editorial standpoint would be to register your disagreement and reject the paper in its present form.

Out of respect for your work and for that of the reviewers, I am willing to extend the end of the discussion one week from today (18 Jan 2021). In case you can solve your differences internally and draft a response to the reviewers with a clear and explicit agreement from all authors and co-authors, I will consider it and decide whether to invite you to submit a revised manuscript. Otherwise the paper will be rejected in its present form, on the grounds that you are unable to answer the reviewers comments in a full and satisfactory manner.

As a final note, I would like to let you know that, before writing this email, I consulted with the other Special Issue guest editors. We all regret this situation. The WALIS Special Issue intends to bring together the paleo sea-level community, bridging across disciplines and ideas, and not creating divisions and unpleasant situations such as this one

Best regards,

Alessio Rovere

---

## Author Comment (AC1) · 31 Jan 2021

There has been a fundamental disagreement between corresponding author and co-authors on one side and handling editor and authors on the other side. I regret to say that there was no solution to the underlying problems and the manuscript had to be withdrawn. As a consequence, the preprint published here does not represent the scientific credo of each individual author and the handling editor was also not happy with it. An updated version of the manuscript which meets the aspiration of the corresponding author is now published elsewhere as a non-peer reviewed preprint.